# Abnormal phase transition between two-dimensional high-density liquid crystal and low-density crystalline solid phases

Wenbin Li[1,2], Longjuan Kong[1,2], Baojie Feng [1], Huixia Fu[1,2], Hui Li[3], Xiao Cheng Zeng [3,4], Kehui Wu[1,2,5] & Lan Chen[1,2]

Some two-dimensional liquid systems are theoretically predicted to have an anomalous phase transition due to unique intermolecular interactions, for example the first-order transition between two-dimensional high-density water and low-density amorphous ice. However, it has never been experimentally observed, to the best of our knowledge. Here we report an entropy-driven phase transition between a high-density liquid crystal and low-density crystalline solid, directly observed by scanning tunneling microscope in carbon monoxide adsorbed on Cu(111). Combined with first principle calculations, we find that repulsive dipole–dipole interactions between carbon monoxide molecules lead to unconventional thermodynamics. This finding of unconventional thermodynamics in two-dimensional carbon monoxide not only provides a platform to study the fundamental principles of anomalous phase transitions in two-dimensional liquids at the atomic scale, but may also help to design and develop more efficient copper-based catalysis.

[1] Institute of Physics, Chinese Academy of Sciences, Beijing 100190, China. [2] School of physics, University of Chinese Academy of Sciences, Beijing 100049, China. [3] Beijing Advanced Innovation Center for Soft Matter Science and Engineering, Beijing University of Chemical Technology, Beijing 100029, China. [4] Department of Chemistry, University of Nebraska Lincoln, Lincoln, NE 68503, USA. [5] Collaborative Innovation Center of Quantum Matter, Beijing 100871, China. Wenbin Li and Longjuan Kong contributed equally to this work. Correspondence and requests for materials should be addressed to H.L. (email: hli@buct.edu.cn) or to K.W. (email: khwu@iphy.ac.cn) or to L.C. (email: lchen@iphy.ac.cn)

A few substances in nature display anomalous density changes, e.g., their density decreases upon transition from liquid to solid. Such anomalous density change may originate either from directional intermolecular hydrogen bonds, e.g., in the case of liquid water[1], or from directional covalent bonds, e.g., in cases of liquid phosphorus[2], gallium[3], and silicon[4,5]. For two-dimensional (2D) matter, e.g., liquid water confined to a nano-slit, a similar anomalous first-order transition between the high-density liquid and low-density amorphous ice has been predicted based on atomistic molecular simulations[1]. However, although a few 2D ices have been experimentally discovered[6,7], the thermally induced phase transition between 2D high-density liquid and low-density solid phase has not been experimentally observed yet, to the best of our knowledge.

Carbon monoxide (CO) adsorption on Cu(111) is an intensively studied system, mainly because copper-based catalysts are widely employed in many important chemical reactions, such as CO oxidation[8,9] and methanol synthesis[10–12]. To date, most studies on CO adsorption have focused on low-coverage regime, where the interactions among CO molecules are negligible. However, real industrial environments usually correspond to high-density adsorption under high CO pressure, where the intermolecular interactions become significant. Unlike most molecular systems where van der Waals or hydrogen bonds are dominant, the dipole moments of CO molecules can create large repulsive forces, and bring in lateral pressure. It is thus of interest to investigate the thermodynamic phenomena in CO/Cu(111) adsorption in the high-density regime.

Here, we report direct experimental observation of an entropy-driven phase transition from 2D high-density liquid crystal (HDLC) to low-density crystalline solid (LDC) in a CO monolayer adsorbed on a Cu(111) surface, in the temperature range 5~77 K. Combined with first-principles calculations, we find that repulsive dipole–dipole interactions among CO molecules leads to the unconventional thermodynamics of 2D CO. This anomalous 2D liquid crystal-to-solid transition may provide a platform that can be investigated with atomic detail for exploring the unusual thermodynamics of 2D matter. In addition, our results could provide a theoretical basis for design and development of more efficient copper-based catalysis.

## Results

**Phase of 2D CO on Cu(111) at 77 K**. First, the adsorption of CO on Cu(111) at a substrate temperature of ~77 K (liquid nitrogen temperature) was studied in our experiments. The scanning tunneling microscopy (STM) measurements of CO on Cu(111) at low coverage ( < 1/3 ML (corresponding to CO molecules on each Cu atom)) reveal no visible molecule (Supplementary Fig. 1(a)), indicating that the CO molecules keep diffusing quickly on the surface in this coverage regime. When the coverage reaches 1/3 ML, a previously reported $(\sqrt{3} \times \sqrt{3})$R30° lattice is formed (Supplementary Fig. 1b)[13–15]. The CO molecules are adsorbed on top sites of underlying next-nearest-neighbor Cu atoms, and each protrusion in STM image represents one CO molecule. However, this coverage is not yet saturated. The 2D density of CO molecules can be further increased by exposing the Cu(111) surface in the CO atmosphere. As a result, the STM images reveal two coexisted ordered structures on Cu(111) surface, as shown in Fig. 1a, b.

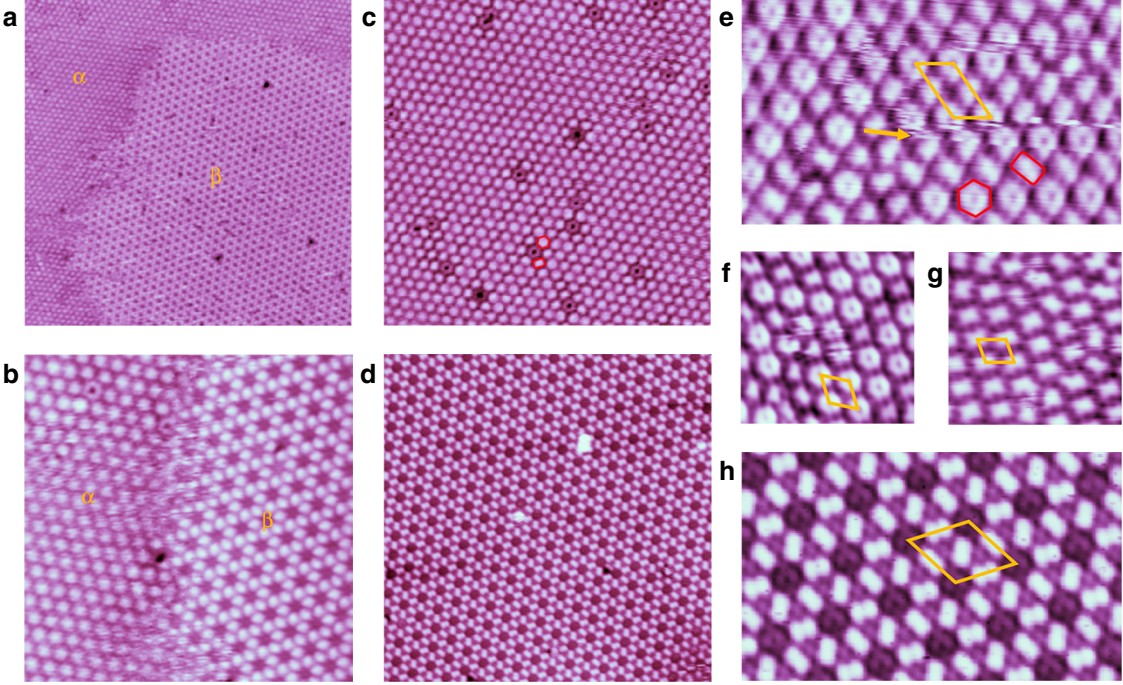

**Fig. 1** STM images of $\alpha$ and $\beta$ phases of 2D CO coexisting on Cu(111) at 77 K. **a** STM image of a large area with coexisted $\alpha$ and $\beta$ phases on Cu(111). **b** STM image of coexisted $\alpha$ and $\beta$ phases with obscure domain boundary, indicating that the $\alpha$ and $\beta$ phases are translated into each other during scanning. **c**, **d** STM images of single domain with pure $\alpha$ and $\beta$ phases, respectively. **e–g** High-resolution STM images of $\alpha$ phase in different areas. The rhomboids represent the unit cells, which correspond to 4 × 7, 4 × 4, and 4 × 3 superlattices with respect to Cu(111), respectively. The two types of protrusions with different shapes are marked by red parallelograms and hexagons in **c** and **e**. **h** High-resolution STM image of $\beta$ phase. The rhombus represent the unit cell of $\beta$ phase with a period of 1.33 nm, corresponding to $(3\sqrt{3} \times 3\sqrt{3})$ R30° with respect to Cu(111). The sizes of the images are: **a** 50.0 × 50.0 nm; **b** 17.0 × 17.0 nm; **c** 19.4 × 19.4 nm; **d** 23.0 × 23.0 nm; **e** 10.8 × 6.4 nm; **f** 5.5 × 5.5 nm; **g** 5.0 × 5.0 nm; **h** 7.6 × 4.6 nm. The scanning parameters are: **a** $V_{tip} = -2.0$ V, I = 31 pA. **b** $V_{tip} = 0.48$ V, I = 196 pA. **c** $V_{tip} = 0.50$ V, I = 195 pA. **d** $V_{tip} = -1.00$ V, I = 90 pA. **e–g** $V_{tip} = -1.00$ V, I = 200 pA. **h** $V_{tip} = -1.00$ V, I = 300 pA

One of the two phases (named $\alpha$ phase) consists of hexagonal closed packed protrusions in large scale (Fig. 1c), which is consistent with previously reported $7 \times 7$ CO superstructure with respect to Cu(111) surface[16]. However, from the close-up STM images (Fig. 1e–g) we found the protrusions actually have two different shapes, a hexagonal one and a parallelogram one (Fig. 1c), respectively, which can transform into each other during scanning (as shown in Fig. 1e). The observed protrusions with various shapes and scales are different from the work reported by B. Wortmann et. al[16]. The random distribution of these two kinds of protrusions result in a few locally ordered superstructures including $4 \times 7$ (Fig. 1e), $4 \times 4$ (Fig. 1f) and $4 \times 3$ (Fig. 1g) with respect to Cu(111), the unit cells, consist of one hexagonal protrusion plus one parallelogram protrusion, one hexagonal protrusion, and one parallelogram protrusion, respectively. It means that the $\alpha$ phase lacks a long-range order, that is the reason why the distances between two nearest protrusions are varied from 0.75 nm to 1.0 nm. Unlike the disordered $\alpha$ phase, the other phase (named $\beta$ phase) adopts a long-range ordered, Kagome-like lattice with three rectangle protrusions in one unit cell (Fig. 1d, h), whose periodicity (1.33 nm) corresponds to $(3\sqrt{3} \times 3\sqrt{3})$ R30° superlattice with respect to Cu(111)[16].

The higher CO density of $\alpha$ phase than the $(\sqrt{3} \times \sqrt{3})$ R30° lattice indicates that the CO ML should involve more closely packed CO clusters. The hexagonal and parallelogram protrusions in $\alpha$ and $\beta$ phases are most likely CO hexamers and tetramers, respectively. A fully relaxed density functional theory (DFT) calculations reveal the stable atomic structures of $4 \times 7$ (Fig. 2a), $4 \times 4$ (Fig. 2b), and $4 \times 3$ lattices (Fig. 2c) of $\alpha$ phase, as well as $(3\sqrt{3} \times 3\sqrt{3})$ R30° lattice of $\beta$ phase (Fig. 2d). It is found the CO molecules in hexamers and tetramers are most likely bounded to the top site of Cu atom. The oxygen atoms in adjacent CO molecules in close-packed clusters stay away from each other, owing to the repulsion between dipole moments of adsorbed CO molecules. The simulated STM images (Fig. 2a–d) of $\alpha$ and $\beta$ phases show topmost bright protrusions with hexagonal or parallelogram pattern perfectly agreeing with the STM images. According to the structural models, the unit cells of $4 \times 7$, $4 \times 4$, and $4 \times 3$ lattices of $\alpha$ phase are constructed of fourteen, six and four CO molecules, respectively, corresponding to coverages of 0.5, 0.38, and 0.33 ML. Considering that most area of $\alpha$ phase is $4 \times 7$ lattice, the total coverage of $\alpha$ phase is estimated to be slightly lower than 0.5 ML. Concurrently, the coverage of $\beta$ phase is 0.48 ML, almost identical to that of $\alpha$ phase.

It is notable the structures of CO ML at 77 K are quite dynamical, where the $\alpha$ and $\beta$ phases can be easily transformed into each other, simply stimulated by the tip bias. A series of successive STM images taken in the same area (Supplementary Fig. 2) clearly illustrate the transformation between the two phases during scanning, reflecting the similar thermal stability and molecular density for the two phases. The nearly identical CO coverage in $\alpha$ and $\beta$ phases in our theoretical model is consistent with the experimental observation of easy and reversible transition between them during STM scanning.

**Phase of 2D CO on Cu(111) at 5 K.** Our major finding was achieved at a lower temperature. When the sample is cooled from 77 K down to 5 K, the coexisting $\alpha$ and $\beta$ phases are transformed into a new distinct phase (named $\gamma$ phase) (Fig. 3a). The $\gamma$ phase consists of perfectly ordered domains separated by domain boundaries. High-resolution STM images (Fig. 3b, c) reveal a honeycomb lattice with 1.0 nm periodicity, corresponding to a $4 \times 4$ superlattice with respect to Cu(111), and there are two round protrusions in each unit cell. Similar to those in $\alpha$ and $\beta$ phases, the bright protrusions in $\gamma$ phase should also correspond

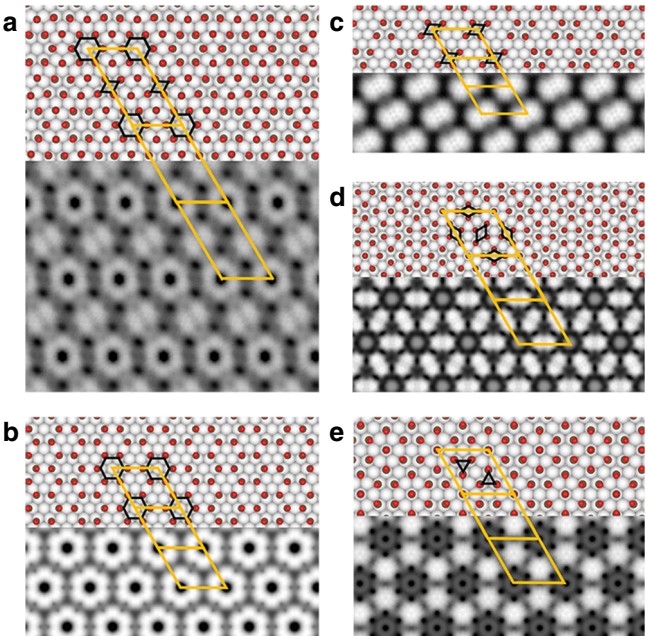

**Fig. 2** Atomic structural models and corresponding STM simulations. **a** $4 \times 7$ superlattice of $\alpha$ phase. **b** $4 \times 4$ superlattice of $\alpha$ phase. **c** $4 \times 3$ superlattice of $\alpha$ phase. **d** $(3\sqrt{3} \times 3\sqrt{3})$R30° superlattice of $\beta$ phase. **e** $4 \times 4$ superlattice of $\gamma$ phase. Red, brown, and gray balls in models represent oxygen, carbon, and copper atoms, respectively. The orange parallelograms and rhombuses represent the unit cells of these phases. The dark blue hexagons, rhombuses, and triangles highlight hexamers, tetramers and trimers of CO molecules, respectively

to CO clusters. However, normal STM images lack structural details of the clusters. In order to resolve the structure of the CO clusters, we functionalized the STM tip by attaching a CO molecule, which has been proven effective to enhance the resolution of STM and Qplus AFM[17–19]. The STM images obtained by a CO-decorated tip are shown in Fig. 3d, e. Each round protrusion in normal STM images can be resolved as a CO trimer, whereas there is an additional CO monomer at the corner of unit cell. Therefore, the atomic model of $\gamma$ phase can be determined as shown in Fig. 2e. The simulated STM image (Fig. 2e) agrees with experimental images shown in Fig. 3b, c obtained by bare tungsten tip very well, further supporting the structure model. Additionally, the domain boundaries are also clearly revealed by CO-decorated tip, and found to consist of CO dimers. The structure models of various domain boundaries are shown in Supplementary Fig. 3. The $4 \times 4$ lattice of the $\gamma$ phase has a CO coverage of 0.44 ML. Furthermore, we counted the numbers of unit cells of pure $\gamma$ phase and its domain boundaries statistically for dozens of STM images of $\gamma$ phase in different areas, and the averaged density of CO molecules in $\gamma$ phase involving domain boundaries was obtained to be 0.43 ML, ~11% lower than 0.48 ML of $\alpha$ and $\beta$ phases.

The fact that the transition from $\alpha/\beta$ phases to $\gamma$ phase is accompanied by a density decreasing (0.48 ML to 0.43 ML) and building-block shrinking (hexamer/tetramer to trimer/dimer), contradicting with the conventional principles of adsorption, where higher coverage is preferred at lower temperature. To understand the mechanism of this anomalous phase transition, we calculated and compared the adsorption energies of different phases, $E_a$, as listed in Table 1. The close adsorption energies of $\alpha$ phase (−0.413, −0.420, and −0.416 eV for the $4 \times 7$, $4 \times 3$, and $4 \times 4$ lattices, respectively) and $\beta$ phase (−0.437 eV) demonstrate close stabilities of $\alpha$ and $\beta$ phases. It is remarkable that the $\gamma$ phase

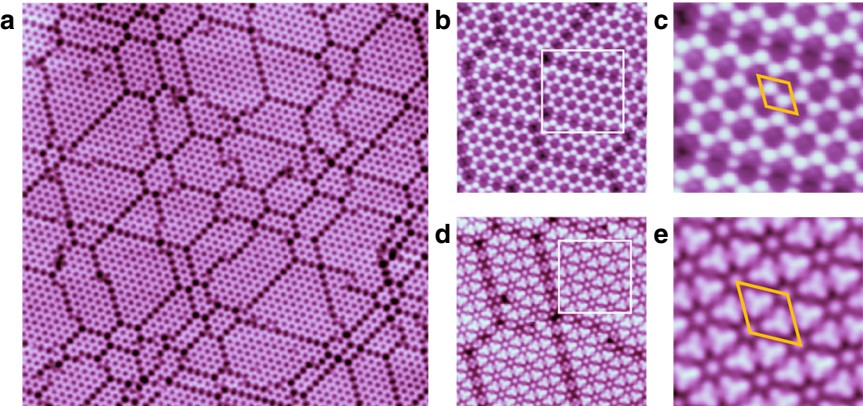

**Fig. 3** STM images of $\gamma$ phase on Cu(111) at 5 K. **a** STM image of $\gamma$ phase of 2D CO at 5 K. **b** High-resolution STM image of $\gamma$ phase obtained by normal W tip. **d** High-resolution STM image of $\gamma$ phase obtained by CO-decorated W tip. **c, e** STM image of areas enlarged from that marked by rectangles in **b** and **d**, respectively. The rhombuses represent units cell of $\gamma$ phase with a period of 1.0 nm, corresponding to 4 × 4 superlattice with respect to Cu(111). The sizes of images are: **a** 50.0 × 50.0 nm; **b** 14.8 × 14.8 nm; **c** 6.0 × 6.0 nm; **d** 10.0 × 10.0 nm; **e** 3.8 × 3.8 nm. The scanning parameters are: a $V_{tip} = -0.8$ V, I = 210 pA. **b, d** $V_{tip} = -1.0$ V, I = 286 pA

| Table 1 Averaged adsorption energy of single CO molecule in various phases | | | | | |
|---|---|---|---|---|---|
| Phase | $\alpha$ | $\beta$ | $\gamma$ | Tetramer | Hexamer |
| Ea (eV/CO) | −0.413 | −0.437 | −0.479 | −0.420 | −0.416 |

has obviously larger formation energy (−0.479 eV) than that of both $\alpha$ and $\beta$ phases, indicating the $\gamma$ phase is indeed more energetically favorable at low temperature. Qualitatively, this can be understood as the adsorption of CO molecules at high coverage creates dense and parallel dipole moments, leading to strong lateral repulsion between adjacent CO molecules. Charge density analysis reveals that the dipole moment of CO molecule is enhanced from 0.24 to 0.27 Debye upon the adsorption on Cu (111) surface, in consistence to previous theoretical study[20]. Such strong repulsion between CO molecules can also be reflected by the tilted angles of CO molecules: 8.5, 10.4, and 13.2 degree for trimer ($\gamma$ phase), tetramer and hexamer ($\alpha$ and $\beta$ phases), respectively. As a result, the higher coverage of CO destabilizes $\beta$ phase than $\gamma$ phase owing to the lateral repulsion.

## Discussion

Although the adsorption energies of CO molecule indicate that the $\gamma$ phase is energetically favorable, the phase transition from low density ($\gamma$ phase) to high density ($\alpha/\beta$ phases) at higher temperature still requires an understanding of the thermodynamics. Usually, the configurational entropy of gas adsorbates can be described by the idea 2D lattice gas (2DLG) model[21],

$$S_{config} = R \ln[(1 - \theta)/\theta] \qquad (1)$$

where $R$ is a constant related with the gas, $\theta$ is the coverage of molecules on surface. Such a model indicates an inverse correlation between $\theta$ and $S_{config}$, which also suggests higher coverage of adsorbed molecules should be preferable at lower temperature, consistent with the Langmuir isotherm[22]. Clearly, our observation contradicts to this idea in the 2DLG model, which is understandable as this model only works when the adsorbate–adsorbate interaction is negligible or slightly repulsive, for example, in the case of $NH_3$ on MgO(100)[23]. The van der Waals isotherm describing a 2D liquid-gas system[24] in which the adsorbate–adsorbate interaction is a weak van der Waals attraction, also indicates that adsorbates at higher coverage are

more stable at lower temperature, for example, hydrogen on Ni(111)[25]. In our case, short intermolecular distances and parallel large dipole moments imply strong CO–CO interactions in the highly dense CO superstructure. Therefore, the $\alpha/\beta$ and $\gamma$ phases of 2D CO layer should have different thermodynamic progressions.

Based on our DFT calculations, isolated CO monomer, dimer, and trimer clusters are all stable on Cu(111) (Supplementary Fig. 4), and the diffusion barriers for CO molecules in all these clusters are nearly identical (~0.06 eV) (Supplementary Fig. 5). In contrast, a single CO hexamer or tetramer is unstable. In other words, the diffusion of CO molecules in isolated tetramers and hexamers is barrierless. The instability of tetramer and hexamer CO clusters, which has been confirmed by our experiments as a transition between tetramers and hexamers in $\alpha$ phase, as well as the transition between $\alpha$ and $\beta$ phases during scanning, indicates that both phases have liquid-like properties. More STM images of the transition in $\alpha$ phase and between $\alpha$ and $\beta$ phases are shown in Supplementary Fig. 6. Combined with the unique orientation of CO molecules and quasi-ordered arrangements, the $\alpha/\beta$ phase at 77 K may be considered as a 2D HDLC phase. Supposing that the repulsive forces from surrounding clusters in $\alpha/\beta$ and $\gamma$ phases are the same, we can qualitatively estimate the diffusion coefficient of $\alpha/\beta$ and $\gamma$ phases through the diffusion barriers of CO molecules in the tetramer/hexamer and trimer, respectively. According to the Arrhenius equation,

$$D = D_0 e^{\frac{-E}{k_B T}} \qquad (2)$$

in which the prefactor ($D_0$) is estimated to be the diffusion coefficient of liquid CO ($D = $ ~$2.5 \times 10^{-5}$ cm²/s at 77 K)[26], the evaluated diffusion coefficient of $\gamma$ phase is ~$2.8 \times 10^{-9}$ cm²/s at 77 K, which is comparable to the value of ice at 252 K ($1.0 \times 10^{-9}$ cm²/s)[27], and is ~$10^{-4}$ times larger than that in $\alpha/\beta$ phase. Therefore, it means that the metastable $\gamma$ phase is a conventional solid, which can be named as a 2D LDC phase.

The solid state of the CO ML obeys the 2D crystal lattice (2DCL) model, in which the entropy of the system is mainly contributed by the vibrations of CO molecules, and the configurational entropy is negligible. According to a previous study[21], the crystal lattice entropy of adsorbed gas molecules in 2DCL model is given as,

$$S_{ad}^0(T) = 0.7 S_{gas}^0(T) - 3.3R \qquad (3)$$

where the superscript 0 refers to the standard pressure of 1 bar, $S_{gas}^0(T)$ is the gas-phase entropy of CO, $R$ is the gas constant, and $T$ is the temperature. Thus, the total entropy of CO/Cu(111) in the solid state is proportional to the coverage $\theta$. That is to say, the $\alpha/\beta$ phase possesses larger entropy than the $\gamma$ phase. Combined with the adsorption energy, the total free energy without configuration entropy can be expressed as,

$$F_a(T) = \theta\left(E_a - S_{ad}^0 T\right) \quad (4)$$

where $\theta$ is the coverage of CO molecules, $E_a$ is the average adsorption energy per CO molecule. Based on this simplified model, we calculated that the $\beta$ phase becomes more stable than the $\gamma$ phase at a temperature higher than 24 K, which agrees well with our experiment. In addition, the configuration entropy becomes more important for the high mobility structure at higher temperature, which can further stabilize the HDLC phase.

Using STM combined with DFT calculations, we discovered an unusual phase transition between HDLC and LDC in a 2D CO layer adsorbed on Cu(111). The anomalous phase transition of CO molecules adsorbed on Cu(111) with high density originates from the balance of Coulombic repulsion between CO molecules and the chemical bonding of CO to Cu(111). This system provides a general insight into the thermodynamics of high-density adsorption of polar molecules on surfaces, which is important in the context of catalyzed chemical reactions in the CO industry. The 2D condensation of molecules may also bring new physics beyond the 2D gas phase, for example, 'molecular graphene'[28,29], molecule cascades[30], and decomposition clusters[31], which relies on the self-assembly of artificial CO molecular crystals. The realization of these artificial structures also requires an understanding of the dynamics of molecules in a high coverage regime. Therefore, the discovery and understanding of this anomalous phase transition not only provides insight into structural phase transitions at the atomic scale, but may also help to improve our capability in controlling surface chemical reactions and functional self-assembly structures for future applications.

## Methods

**Preparation and characterization of samples**. Our experiments were carried out in a home-built low-temperature STM/MBE system with a base pressure of $6 \times 10^{-11}$ Torr. Single-crystal Cu(111) was cleaned by cycles of argon ion sputtering ($2 \times 10^{-5}$ Torr, 1 kV) and annealing at 950 K. The $\alpha$ and $\beta$ phases of 2D CO was prepared by dosing CO on Cu(111) at LN$_2$ temperature in UHV chamber with the pressure of $1.0 \times 10^{-10}$ Torr (CO partial pressure is $4 \times 10^{-11}$ Torr) >48 h. The $\gamma$ phase on Cu(111) surface is obtained by cooling sample with $\alpha$ and $\beta$ phases down to LHe temperature directly. The STM images of $\alpha$ and $\beta$ phases were taken at 77 K, whereas that of $\gamma$ phase were taken at 5 K. All the STM data were analyzed and rendered using WSxM software[32].

**Theoretical calculations and STM simulations**. The first-principles calculations have been carried out in the framework of DFT using Vienna Ab initio Simulation Package (VASP)[33]. The projector-augmented wave pseudopotentials with a plane-wave basis set were utilized to calculate the interaction between the valence electrons and the core electrons. The plan-wave cutoff energy was set to 500 eV. Four periodic layers of Cu were employed to model the Cu(111) surface, where the two upmost layers and CO molecules were relaxed and the bottom two layers of Cu atoms were fixed as bulk position. In order to exclude periodic surface–surface interactions, a vacuum region of $\geq 15$ Å was chosen. A $2 \times 2 \times 1$ Monkhorst-Pack k-mesh was used to sample the 2D Brillouin Zone of $\alpha$ phase with a $4 \times 7$ in-plane supercell, whereas the density of k-mesh keeps for other phases with different supercell.

It is noted that the standard DFT with the local or generalized gradient density approximations fails to predict the correct adsorption site for CO on metal surfaces. For instances, several previous theoretical calculations predict that the stable adsorption site of CO on the Cu(111) or Pt(111) at low coverage is the hollow site[34–39], whereas the top site with the carbon end down is preferred from the experimental point of view[15,40–42]. So far, several theoretical attempts have been devoted to correctly predict the favorite adsorption site, ensuring the top site to be preferred as found experimentally[38,39,43–45]. One of the most popular options is the molecular DFT+U method suggested by Kresse and co-workers[44]. In this way, the calculated site preference and adsorption energy reproduce the experimental results. Similarly, the PBE+U approach with $U = 6$ eV for C and O

atoms was used to restore the correct prediction that the top site adsorption is preferred on Cu(111), in accordance with experiment[15,31,40]. We also employ, for comparison, the same PBE+U method with the empirical parameter $U = 6$ eV to present an extensive DFT investigation of the adsorption of CO on Cu (111) surface. In order to further verify the PBE+U approach, we study the adsorption of one CO molecule in a $4 \times 4$ supercell on two adsorption sites: top site, where CO is sitting vertically above a metal atom in the top layer, and the hcp hollow site, where CO is sitting vertically above a metal atom in the second topmost layer. The calculated energy difference between the hcp and top site is $\sim -0.013$ eV, in agreement with the value of Ref. [31].

**Data availability**. All data and related analyses are available from the corresponding authors upon request.

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

## Acknowledgements

The authors thank Professor Sheng Meng from IOP, CAS for useful discussions. This work was supported by the MOST of China (grants nos. 2016YFA0300904, 2016YFA0202301, 2013CBA01601, 2013CB921702), the NSF of China (grants nos. 11674366, 11674368, 11334011, 11304368, 11374333), the Strategic Priority Research Program of the Chinese Academy of Sciences (grant no. XDB07020100, XDPB06), and the BUCT Fund for Disciplines Construction (Project no. XK1702).

## Author contributions

L.C. and K.W. designed the experiments; W.L. and B.F. performed experiments as well as data analysis under the supervision of L.C.; L.K. and H.F. performed the DFT calculations as well as data analysis under the supervision of H.L.; L.C., H.L., X.C.Z., and K.W. wrote the manuscript with contribution from all the authors; all co-authors contributed to data analyses and discussions.

## Additional information

**Competing interests:** The authors declare no competing financial interests.

