## [Peer Review File · Nature Communications]

Reviewer #1 (Remarks to the Author):

The manuscript by Wenbin Li et. al. reported an direct observation of entropy-driven phase transition between high density liquid and low density solid using scanning tunneling microscope. It provides insight into the structural phase transitions at atomic scales and may also help to improve the capability in controlling surface chemical reactions and functional self-assembly structure for future applications. It is an interesting experimental work with substantial theoretical supporting. However, the manuscript needs some clarification and improvement before it can be accepted for publication.

1. In page3, "from the close-up STM image (Fig. 1(e-g)), we found the protrusions actually have two different shapes, a hexagonal one and a rectangle one (Fig. 1(c)". Does the rectangle mean a strict rectangle or just a parallelogram? I think it is better to mark the hexagonal one and the rectangle one in figure 1(c).

2. In page4, "The nearly identical CO coverage in α and β phases in our theoretical model nicely explains the experimental observation of the co-existence of these two phases at 77 K, as well as the easily reversible transition between them during STM scanning, which in turn supports our theoretical models." The theoretical support of the nearly identical CO coverage in α and β phases is not strong enough to explain the experimental observation of the co-existence of these two phases at 77 K and the easily reversible transition between them.

3. In page4, "The simulated STM image (Fig. 3(e)) agrees with experimental images very well" in fig.3(e), the simulated STM image has dark feature in the vertex of the unit cell but the corresponding STM image feature is bright, is it possible to explain why?

4. In page5, "If the domain boundaries are further involved, the coverage of the γ phase is estimated to be 0.43 ML", how the domain boundaries be further involved?

5. In page6, "The instability of tetramer and hexamer CO clusters that have been confirmed by our experiment as transition between tetramers and hexamers in α phase as well as the transition between α and β phases during scanning, indicates both phases have liquid-like properties." α and β phases have liquid-like properties, it is better to give more evidence to support that it is liquid since this is important to say that the reported phase transition is between high density liquid and low density solid.

6. There are many obvious typos and grammatical errors in the manuscript, the authors should go through very carefully, e.g., "idea", "ase", "bD",...

Reviewer #2 (Remarks to the Author):

The paper is very interesting and touches on a subject (entropy) not usually considered in studying condensed phases, like adsorbates on crystalline substrates. The author's observations are quite nice and bring new physical insights into the forces determining phase transitions.

As minor points the authors should perhaps have a colleague with better knowledge of English to help in the presentation

Reply to referee 1

C1: The manuscript by Wenbin Li et. al. reported an direct observation of entropy-driven phase transition between high density liquid and low density solid using scanning tunneling microscope. It provides insight into the structural phase transitions at atomic scales and may also help to improve the capability in controlling surface chemical reactions and functional self-assembly structure for future applications. It is an interesting experimental work with substantial theoretical supporting. However, the manuscript needs some clarification and improvement before it can be accepted for publication.

R1: We thank the reviewer for his positive comments to our work. We Address all the questions below:

C2: In page3, “from the close-up STM image (Fig. 1(e-g)), we found the protrusions actually have

two different shapes, a hexagonal one and a rectangle one (Fig. 1(c)). Does the rectangle mean a

strict rectangle or just a parallelogram? I think it is better to mark the hexagonal one and the rectangle one in figure 1(c).

R2: We thank the reviewer for pointing out our mistake. The word “rectangle” does not mean a strict rectangle, but just a parallelogram. In order to avoid misunderstanding, we replaced “rectangle” with “parallelogram” in revised manuscript. And following the reviewer’s suggestion, we also mark the hexagonal one and the parallelogram one in Fig. 1(c) and (e) in revised manuscript.

C3: *In page4, "The nearly identical CO coverage in α and β phases in our theoretical model nicely explains the experimental observation of the co-existence of these two phases at 77 K, as well as the easily reversible transition between them during STM scanning, which in turn supports our theoretical models." The theoretical support of the nearly identical CO coverage in α and β phases is not strong enough to explain the experimental observation of the co-existence of these two phases at 77 K and the easily reversible transition between them.*

R3: We thank the reviewer for this insightful comment. We fully agree that an identical coverage of CO simply means no additional CO molecule is needed during a possible phase transition. However, it is not strong enough to support the "easy" transition between these two phases. On the other hand, the close binding energies and low diffusion barrier of CO molecule based on the DFT calculations can indeed support the coexistence and easy transition between the α and β phases.

To avoid misunderstanding, we have made corresponding modifications in the manuscript: "The nearly identical CO coverage in α and β phases in our theoretical model is consistent with the experimental observation of easy and reversible transition between them during STM scanning."

C4: *In page 4, "The simulated STM image (Fig. 3(e)) agrees with experimental images very well" in fig.3(e), the simulated STM image has dark feature in the vertex of the unit cell but the corresponding STM image feature is bright, is it possible to explain why?*

R4: We are sorry that this point has not been stated clearly enough in our manuscript. The Fig. 2(b-c) and Fig. 2(d-e) were obtained with different tips: Fig. 2(b-c) were obtained by a bare tungsten tip, while Fig. 2(d-e) were obtained by CO -decorated tip. Actually, the simulated STM image (Fig. 3(e)) agrees with experimental images of Fig. 2(b, c) (with a tungsten tip) very well, but it does not agree with those obtained by a CO-decorated tip (Fig. 2(d-e)). As we know, the contrast in an STM image corresponds to the convolution of the electronic states of both tip and sample. Due to the spatial flat electronic states of the tungsten tip, the brightness of STM images obtained by a W tip is contributed mainly by the density of states (DOS) of sample. Therefore, the simulated STM images calculated from the DOS of the sample can explain the experimental images by a tungsten tip. One can see that both experimental STM images (Fig. 2(b, c) by tungsten tip and simulated image (Fig. 3(e)) have dark features in the vertex of the unit cell.

On the other hand, according to previous studies (L. Bartels *et al.* Appl. Phys. Lett. 71, 213 (1997); D. Dracova *et al.* Int. J. Quantum Chem. 106, 1419 (2006)), a single CO molecule on Cu(111) exhibits a dark dot on the surface with a bright ring around, while the contrast of CO molecule is reversed when using a CO functionalized tip. Therefore, we can use a CO- decorated tip to identify a CO molecule in the vertex of unit cell as a bright dot. We cannot use the same way to recognize the clusters in α/β phase because the CO molecule cannot be stably adsorbed on the tip at 77K.

In the revised manuscript, we have made corresponding modifications: “The simulated STM image (Fig. 3(e)) agrees with experimental images Fig.2(b) and (c) obtained by bare tungsten tip very well”.

C5: *In page5, “If the domain boundaries are further involved, the coverage of the γ phase is estimated to be 0.43 ML”, how the domain boundaries be further involved?*

R5: The calculation of the coverage of γ phase involving domain boundary is as follow. The atomic structural models of pure γ phase and its domain boundary are known as shown in Fig. 3(e) in manuscript and Fig. S3(i) in supplemental materials, respectively. Thus we know how many CO molecules in one unit cell of pure γ phase and its domain boundary separately based on the models. Then we counted the numbers of unit cells of pure γ phase and its domain boundary statistically in an STM images of γ phase with $50\text{ nm} \times 50\text{ nm}$ similar like Fig. 2(a). We statistically analyzed dozens of STM images on different areas, and obtained the average number of CO molecules on a certain area of surface. The coverage of γ phase with domain boundary involved was estimated to be 0.43 ML.

To make it more clear, we added the above details into the revised manuscript: “Furthermore, we counted the numbers of unit cells of pure γ phase and its domain boundaries statistically for dozens of STM images of γ phase in different areas, and the averaged density of CO molecules in γ phase involving domain boundaries was obtained to be 0.43 ML.”

C6: *In page6, “The instability of tetramer and hexamer CO clusters that have been confirmed by our experiment as transition between tetramers and hexamers in α phase as well as the transition between α and β phases during scanning, indicates both phases have liquid-like properties.” α and β phases have liquid-like properties, it is better to give more evidence to support that it is liquid since this is important to say that the reported phase transition is between high density liquid and low density solid.*

R6: We thank the reviewer for his suggestion. Actually, in our calculations, a single CO hexamer or tetramer is unstable. In another word, the diffusion of CO molecules in tetramer and hexamer is barrierless, suggesting the α and β phases have liquid-like properties.

From the experimental side, the closed up STM images (Fig. 1(e-g) in manuscript) of α phase show that it has only short range order, but without a long-range order. Moreover, most of hexamers and tetramers are continuously moving at any moment, which are the characters of liquid. To strength this point, we show the STM images of α phase in the same area but taken at different time, as shown in Fig. R(c) and (d). The arrangement of tetramers and hexamers are disordered, and the CO clusters are changed and moving, suggesting a liquid-like character.

At last, we found the α phase and β phase can be easily transitioned to each other during scanning, as shown in Fig.S2 in supplemental materials. We give the STM images in Fig. R(a) and (b) showing the CO molecules at the boundary between α phase and β phase are moving and

changing all the time.

The barrierless diffusion, lacking long-rang order, and continuous movement of CO molecules on the surface show it clearly that the α and β phases should be regard as liquid crystal rather than solid.

The STM images of Fig. R(a-d) as the evidences that α and β phases are liquid crystal has been added into supplemental materials of manuscript.

Fig. R. (a, b) STM images taken on the area with boundaries separating α and β phases. (c, d) High resolution STM images of α phase in the same area but taken at different time. Examples of clusters transformation are marked by colors cycles. The scanning parameters are: (a) $V_{\text{tip}} = -500$ mV, $I = 301$ pA; (b) $V_{\text{tip}} = -477$ mV, $I = 195$ pA; (c, d) $V_{\text{tip}} = -1.00$ V, $I = 207$ pA.

C7: There are many obvious typos and grammatical errors in the manuscript, the authors should go through very carefully, e.g., “idea”, “ase”, “bD”

R7: We thank the reviewer again. We try our best to revise the typos and grammatical errors in manuscript carefully.

Reply to referee 2:

C: The paper is very interesting and touches on a subject (entropy) not usually considered in studying condensed phases, like adsorbates on crystalline substrates. The author's observations are quite nice and bring new physical insights into the forces determining phase transitions.

As minor points the authors should perhaps have a colleague with better knowledge of English to help in the presentation

R: We thank the reviewer for his high evaluation on our manuscript. In revised manuscript, we try our best to the English presentation in revised manuscript.

Reviewer #1 (Remarks to the Author):

The questions I raised are more or less addressed. No further comments.

Reply to referee 1

C1: The questions I raised are more or less addressed. No further comments.

R1: Thank very much for reviewer's comments.